# Type 2 Diabetes-Related Variants Influence the Risk of Developing Prostate Cancer: A Population-Based Case-Control Study and Meta-Analysis

**DOI:** 10.3390/cancers14102376

**Published:** 2022-05-12

**Authors:** José Manuel Sánchez-Maldonado, Ricardo Collado, Antonio José Cabrera-Serrano, Rob Ter Horst, Fernando Gálvez-Montosa, Inmaculada Robles-Fernández, Verónica Arenas-Rodríguez, Blanca Cano-Gutiérrez, Olivier Bakker, María Inmaculada Bravo-Fernández, Francisco José García-Verdejo, José Antonio López López, Jesús Olivares-Ruiz, Miguel Ángel López-Nevot, Laura Fernández-Puerta, José Manuel Cózar-Olmo, Yang Li, Mihai G. Netea, Manuel Jurado, Jose Antonio Lorente, Pedro Sánchez-Rovira, María Jesús Álvarez-Cubero, Juan Sainz

**Affiliations:** 1Genomic Oncology Area, GENYO, Centre for Genomics and Oncological Research, Pfizer/University of Granada/Andalusian Regional Government, PTS Granada, 18016 Granada, Spain; josemanuel.sanchez@genyo.es (J.M.S.-M.); antonio.cabrera@genyo.es (A.J.C.-S.); inmaculadarobles@gmail.com (I.R.-F.); veronica.arenas@genyo.es (V.A.-R.); manuel.jurado.sspa@juntadeandalucia.es (M.J.); jose.lorente@genyo.es (J.A.L.); mjesusac@ugr.es (M.J.Á.-C.); 2Hematology Department, Virgen de las Nieves University Hospital, 18012 Granada, Spain; laurafdezpuerta@gmail.com; 3Instituto de Investigación Biosanataria IBs. Granada, 18012 Granada, Spain; 4Medical Oncology Department, Hospital de San Pedro Alcántara, 10003 Cáceres, Spain; ricardo.collado@salud-juntaex.es (R.C.); inmaculada.bravo@salud-juntaex.es (M.I.B.-F.); jesus.olivares@salud-juntaex.es (J.O.-R.); 5Department of Internal Medicine and Radboud Centre for Infectious Diseases, Radboud University Nijmegen Medical Center, 6525 GA Nijmegen, The Netherlands; rterhorst@cemm.oeaw.ac.at (R.T.H.); yang.li@helmholtz-hzi.de (Y.L.); mihai.netea@radboudumc.nl (M.G.N.); 6Department of Medical Oncology, Complejo Hospitalario de Jaén, 23007 Jaén, Spain; fernando.galvez.sspa@juntadeandalucia.es (F.G.-M.); francisco.garcia.verdejo.sspa@juntadeandalucia.es (F.J.G.-V.); josea.lopez.l.sspa@juntadeandalucia.es (J.A.L.L.); pedro.sanchez.rovira.sspa@juntadeandalucia.es (P.S.-R.); 7Department of Biochemistry and Molecular Biology III, Faculty of Medicine, University of Granada, 18016 Granada, Spain; blanca.cano.sspa@juntadeandalucia.es; 8Department of Genetics, University Medical Center Groningen, University of Groningen, 9713 GZ Groningen, The Netherlands; o.b.bakker@umcg.nl; 9Immunology Department, Virgen de las Nieves University Hospital, 18012 Granada, Spain; manevot@ugr.es; 10Urology Department, Virgen de las Nieves University Hospital, 18012 Granada, Spain; josem.cozar.sspa@juntadeandalucia.es; 11Centre for Individualised Infection Medicine (CiiM) & TWINCORE, Joint Ventures between the Helmholtz-Centre for Infection Research (HZI) and the Hannover Medical School (MHH), 30625 Hannover, Germany; 12Department for Immunology & Metabolism, Life and Medical Sciences Institute (LIMES), University of Bonn, 53115 Bonn, Germany; 13Department of Medicine, Faculty of Medicine, University of Granada, 18016 Granada, Spain; 14Department of Legal Medicine, Faculty of Medicine, University of Granada, 18016 Granada, Spain; 15Department of Biochemistry and Molecular Biology I, Faculty of Sciences, University of Granada, 18071 Granada, Spain

**Keywords:** prostate cancer, genetic susceptibility, type 2 diabetes-related variants

## Abstract

**Simple Summary:**

We investigated the influence of GWAS-identified variants for T2D in modulating prostate cancer (PCa) risk through a meta-analysis of our data with those from the UKBiobank and FinnGEn cohorts and four large European cohorts. We found that genetic variants within the *FTO*, *HNF1B*, and *JAZF1* loci were associated with PCa risk. Our results also suggested, for the first time, a potentially interesting association of SNPs within *NOTCH2* and *RBMS1* genes that need to be further explored and validated. This study also shed some light onto the functional mechanisms behind the observed associations, and demonstrated that the *HNF1B*_rs7501939_ polymorphism correlated with lower levels of SULT1A1, an enzyme responsible for the sulfate conjugation of multiple endogenous and exogenous compounds. Furthermore, we found that SNPs within the *HFN1B*, *NOTCH2*, and *RBMS1* genes impacted PCa risk through the modulation of mRNA gene expression levels of their respective genes. However, given the healthy nature of the subjects included in the cohort used for functional experiments, the link between the *HNF1B* locus and SULT1A1 should be considered still speculative and, therefore, requires further validation.

**Abstract:**

In this study, we have evaluated whether 57 genome-wide association studies (GWAS)-identified common variants for type 2 diabetes (T2D) influence the risk of developing prostate cancer (PCa) in a population of 304 Caucasian PCa patients and 686 controls. The association of selected single nucleotide polymorphisms (SNPs) with the risk of PCa was validated through meta-analysis of our data with those from the UKBiobank and FinnGen cohorts, but also previously published genetic studies. We also evaluated whether T2D SNPs associated with PCa risk could influence host immune responses by analysing their correlation with absolute numbers of 91 blood-derived cell populations and circulating levels of 103 immunological proteins and 7 steroid hormones. We also investigated the correlation of the most interesting SNPs with cytokine levels after in vitro stimulation of whole blood, peripheral mononuclear cells (PBMCs), and monocyte-derived macrophages with LPS, PHA, Pam3Cys, and *Staphylococcus Aureus*. The meta-analysis of our data with those from six large cohorts confirmed that each copy of the *FTO*_rs9939609A_, *HNF1B*_rs7501939T_, *HNF1B*_rs757210T_, *HNF1B*_rs4430796G_, and *JAZF1*_rs10486567A_ alleles significantly decreased risk of developing PCa (*p* = 3.70 × 10^−5^, *p* = 9.39 × 10^−54^, *p* = 5.04 × 10^−54^, *p* = 1.19 × 10^−71^, and *p* = 1.66 × 10^−18^, respectively). Although it was not statistically significant after correction for multiple testing, we also found that the *NOTCH2*_rs10923931T_ and *RBMS1*_rs7593730_ SNPs associated with the risk of developing PCa (*p* = 8.49 × 10^−4^ and 0.004). Interestingly, we found that the protective effect attributed to the *HFN1B* locus could be mediated by the SULT1A1 protein (*p* = 0.00030), an arylsulfotransferase that catalyzes the sulfate conjugation of many hormones, neurotransmitters, drugs, and xenobiotic compounds. In addition to these results, eQTL analysis revealed that the *HNF1B*_rs7501939_, *HNF1B*_rs757210_, *HNF1B*_rs4430796_, *NOTCH2*_rs10923931_, and *RBMS1*_rs7593730_ SNPs influence the risk of PCa through the modulation of mRNA levels of their respective genes in whole blood and/or liver. These results confirm that functional TD2-related variants influence the risk of developing PCa, but also highlight the need of additional experiments to validate our functional results in a tumoral tissue context.

## 1. Introduction

Prostate cancer (PCa) is the second most common cancer worldwide and one of the first leading causes of cancer-related deaths in men in developed countries [1]. It accounts for 7.3% of all cancers, with an incidence of 37.5 per 100,000 individuals [2]. Despite the refinements in prevention strategies, a total of 1.4 million new cases were diagnosed in 2020 [2], and the incidence of the disease is increasing over the world, likely due to the interaction of both inherited and modifiable factors [3,4]. 

Although PCa has a high prevalence among males, only age, family history, and ethnicity have been established as major risk factors for the disease, with an attributable effect ranging from 5 to 9% of the cases [5]. In addition, rare highly penetrant mutations in specific genes, high levels of endogenous androgens, smoking, alcohol consumption, exposure to chemical compounds, sexually transmitted infections, diet, obesity, insulin-like growth factors, and type 2-diabetes (T2D) have been suggested as important modulators of the prostatic tumorigenesis [6]. Among the modifiable risk factors, T2D has attracted significant attention, since it has been consistently identified as a protective factor for PCa development [7,8,9], but it seems to induce disease progression. Several studies have suggested that the use of anti-diabetic drugs such as metformin might account for this protective effect of T2D on PCa risk [10,11], but other studies were not able to confirm these results in larger cohorts [12], which suggested that the protective effect attributed to T2D on PCa depend on common molecular pathways between these traits rather than the use of anti-diabetic drugs.

Although research on the specific pathways interfering in the development of T2D and PCa traits is still under way, a mounting body of evidence suggests that these diseases might share a genetic component [13,14,15,16]. Recent studies have reported that common genetic polymorphisms within T2D-related genes have an important role in modulating the risk of many cancers [17,18,19], but, so far, only a few studies have investigated the impact of diabetogenic variants on PCa risk, showing controversial results [20,21,22,23]. Considering this background, we decided to conduct a population-based case-control study including 994 subjects (304 PCa patients and 686 controls) to evaluate whether 57 diabetogenic variants identified through genome-wide association studies (GWAS) are associated with the risk of developing Pca. In order to validate the association of T2D-related markers with Pca risk, when possible, we performed a meta-analysis with data from previous genetic studies. Finally, we analyzed whether the most interesting markers correlated with absolute numbers of 91 blood-derived cell populations, 106 immunological serum proteins, 7 steroid hormones, and 9 cytokines (IFNγ, IL1β, IL1Ra, IL6, IL8, IL10, IL17, IL22, and TNFα) after stimulation of whole blood, peripheral blood mononuclear cells (PBMCs), and monocyte-derived macrophages with LPS, PHA, Pam3Cys, and *Staphylococcus aureus*. 

## 2. Materials and Methods

### 2.1. Study Population

The study cohort consisted of 304 Caucasian PCa patients and 686 male healthy controls recruited in the Virgen de las Nieves University hospital (Granada, Spain) and the Complejo Hospitalario de Cáceres (Cáceres, Spain). Only patients without any prior history of malignancy, and who were not treated before blood withdrawal, were enrolled in this study. Patient characteristics are included in Table 1. The diagnosis of PCa was assigned by physicians, and fulfilled the international criteria [24]. Male controls with a mean age of 58.92 were blood donors from the Centro Regional de Transfusiones Sanguíneas de Granada (CRTS) and were selected from the same geographical region of the cases. In accordance with the Declaration of Helsinki, all participants gave their written informed consent to participate in the study and the ethical committees of the participant institutions approved the study.

### 2.2. SNP Selection and Genotyping

An extensive literature search concerning the mechanism of action of T2D-related genes was performed to select candidate genes that might affect the risk of developing PCa. SNPs were assessed on the basis of NCBI data, and were selected according to their known or putative functional consequences, i.e., their modifying influence on the structure of proteins, transcription level, or alternative splicing mechanisms. In total, 57 SNPs in 49 genes were selected for this study (Table 2).

Selected variants for T2D were genotyped using KASPar^®^ assays (LGC Genomics, London, UK) according to the manufacturer’s instructions. For internal quality control, 5% of samples were randomly included as duplicates. Concordance between the original and the duplicate samples for the 57 SNPs was ≥99.0%. Call rates for all SNPs were ≥90.0%.

### 2.3. Statistical Analysis

The Hardy–Weinberg Equilibrium (HWE) tests were performed in the control group by a standard observed–expected chi-squared (χ^2^) test. Logistic regression analyses were used to assess the effects of the genetic polymorphisms on PCa risk using dominant, recessive, and log-additive models of inheritance. Overall analyses were adjusted for age, and conducted using Stata (v12.1). Statistical power was calculated using the Quanto software (vs. 12.4; log-additive model). 

In order to account for multiple testing, we calculated an adjusted significance level using data from the SNPclip Tool (https://ldlink.nci.nih.gov/?tab=snpclip, accessed on 8 May 2020), which consider the number of independent marker loci (*n* = 52). Given the high correlation between the log-additive and dominant models of inheritance, we corrected by log-additive and recessive models, resulting in a significant threshold for the main effect analysis of 0.00048 (0.05/52 SNPs/2 inheritance models). Since a study-wide significance threshold considering all these factors is generally perceived as a too conservative test, we also assessed the magnitude of observed associations between selected SNPs and risk of PCa through a quantile–quantile (QQ) plot generated from the results of the study population. The observed association *p*-values were ranked in order from smallest to largest on the y-axis and plotted against the expected results from a theoretical ~χ^2^-distribution under the null hypothesis of no association on the x-axis. 

### 2.4. Meta-Analysis

With the aim of assessing the consistency of the association between T2D-related SNPs and the risk of developing PCa, we performed a meta-analysis of our data with those from publicly available GWAS. We downloaded association estimates from the PheWeb site (https://pheweb.sph.umich.edu/, accessed on 11 May 2020) for 6311 PCa cases; 74,685 controls from the FinnGen research project; and 5993 PCa cases and 168,999 controls from the UK Biobank project (UKBiobank TOPMed-imputed). Details on genome-wide associations have been previously reported [70]. Briefly, analyses on binary outcomes were conducted using the SAIGE generalized mixed logistic regression model, adjusting for genetic relatedness, sex, birth year, and the first four principal components. For White British participants of the UK Biobank, endpoint definitions were generated from electronic health-records-derived ICD billing codes, and endpoint definitions for the FinnGen data can be found at risteys.finngen.fi (Risteys = intersection in Finnish). We also validated the association of genetic markers using data from previously published studies that were selected according to the following criteria: (1) GWAS or candidate-gene association studies found in PUBMED (https://www.ncbi.nlm.nih.gov/pubmed, accessed on 13 May 2020) using the following key words: prostate cancer, case-control association study, type 2 diabetes, genetic polymorphisms; (2) Studies using Caucasian populations; (3) Availability of association estimates according to a log-additive model of inheritance; (4) Hardy–Weinberg equilibrium in the control group; and (5) Written in English (Figure 1). We pooled the Odds Ratios (ORs) using a fixed-effect model. Coefficients with a *p*-value *≤* 0.05 were considered significant. I^2^ statistic was used to assess heterogeneity between studies. All statistics were calculated using STATA (v. 12).

### 2.5. cQTL Analysis of the T2D-Related Variants

Cytokine stimulation experiments were conducted in the 500 Functional Genomics (500FG) cohort from the Human Functional Genomics Project (HFGP; http://www.humanfunctionalgenomics.org/, accessed on 7 July 2020). The HFGP study was approved by the Arnhem-Nijmegen Ethical Committee (no. 42561.091.12), and biological specimens were collected after informed consent was obtained. We investigated whether any of the 57 T2D-related SNPs correlated with cytokine levels (IFNγ, IL1β, IL1Ra, IL6, IL8, IL10, IL17, IL22, and TNFα) after the stimulation of peripheral blood mononuclear cells (PBMCs), macrophages, or whole blood from 172 healthy men with LPS (1 or 100 ng/mL), PHA (10 μg/mL), Pam3Cys (10 μg/mL), and *Staphylococcus aureus*. After log transformation, linear regression analyses adjusted for age were used to determine the correlation of selected SNPs with cytokine expression quantitative trait loci (cQTLs). All analyses were performed using R software (http://www.r-project.org/, accessed on 8 May 2020). In order to account for multiple comparisons, we used a significant threshold of 0.000106 (0.05/52 independent SNPs × 9 cytokines).

Details on PBMCs isolation, macrophage differentiation, and stimulation assays have been reported elsewhere [72,73,74]. Briefly, PBMCs were washed twice in saline, and suspended in medium (RPMI 1640) supplemented with gentamicin (10 mg/mL), L-glutamine (10 mM), and pyruvate (10 mM). PBMC stimulations were performed with 5 × 10^5^ cells/well in round-bottom 96-well plates (Greiner) for 24 h in the presence of 10% human pool serum at 37 °C and 5% CO_2_. Supernatants were collected and stored in −20 °C until used for ELISA. LPS (100 ng/mL), PHA (10 μg/mL), and Pam3Cys (10 μg/mL) were used as stimulators for 24 or 48 h. Whole blood stimulation experiments were conducted using 100 μL of heparin blood that was added to a 48-well plate and subsequently stimulated with 400 μL of LPS and PHA (final volume 500 μL) for 48 h at 37 °C and 5% CO_2_. Supernatants were collected and stored in −20 °C until used for ELISA. Concentrations of human cytokines were determined using specific commercial ELISA kits (PeliKine Compact, Amsterdam, The Netherlands or R&D Systems, Minneapolis, MN, USA), following the manufacturer’s instructions. 

### 2.6. Correlation between T2D-Related Polymorphisms and Cell Counts of 91 Blood-Derived Immune Cell Populations and 103 Serum/Plasmatic Immunological Proteins

We also investigated whether selected polymorphisms had an impact on blood cell counts by analyzing a set of 91 manually annotated immune cell populations and genotype data from the 500 FG cohort that consisted of 172 healthy men (Appendix A). Cell populations were measured by 10-color flow cytometry (Navios flow cytometer, Beckman Coulter) after blood sampling (2–3 h), and cell count analysis was performed using the Kaluza software (Beckman Coulter, v. 1.3). In order to reduce inter-experimental noise and increase statistical power, cell count analysis was performed by calculating parental and grandparental percentages, which were defined as the percentage of a certain cell type within the cell populations one or two levels higher in the hierarchical definitions of cell sub-populations [75]. Detailed laboratory protocols for cell isolation, reagents, gating, and flow cytometry analysis have been reported elsewhere [76], and the accession number for the raw flow cytometry data and analyzed data files are available upon request to the authors (http://hfgp.bbmri.nl, accessed on 8 May 2020). A proteomic analysis was also performed in serum and plasma samples from the 500 FG cohort. Circulating proteins were measured using the commercially available Olink^®^ Inflammation panel (Olink, Sweden), which resulted in the measurement of 103 different biomarkers (Appendix A Appendix A). Proteins levels were expressed on a log2-scale as normalized protein expression values, and normalized using bridging samples to correct for batch variation. Considering the number of proteins (*n* = 103) and cell populations (*n* = 91) tested, *p*-values of 9.33 × 10^−6^ and 1.05 × 10^−5^ were set as significant thresholds for the proteomic and cell-level variation analysis, respectively.

### 2.7. Correlation between Steroid Hormone Levels and T2D-Related SNPs 

We also measured serum levels of seven steroid hormones (androstenedione, cortisol, 11-deoxy-cortisol, 17-hydroxy progesterone, progesterone, testosterone, and 25 hydroxy vitamin D3) in the 500 FG cohort. Complete protocol details of steroid hormone measurements have been reported elsewhere [74]. Hormone levels and genotyping data were available for a total of 167 subjects. After log-transform, correlation between steroid hormone levels and T2D-related SNPs was evaluated by linear regression analysis adjusted for age. In order to avoid a possible bias, we excluded from the analysis those subjects that were using oral contraceptives, or those subjects in which this information was not available. The significance threshold was set to 0.000137 considering the number of independent SNPs tested (*n* = 52) and the number of hormones determined (*n* = 7).

### 2.8. In Silico Functional Analysis

Once we assessed the correlation of T2D-related SNPs with cytokine and steroid hormone levels, we used the HaploReg SNP annotation tool to further investigate the functional consequences of each specific variant (http://www.broadinstitute.org/mammals/haploreg/haploreg.php, accessed on 8 July 2020). We also assessed whether any of the potentially interesting markers correlated with mRNA expression levels of their respective genes using data from public eQTL browsers (GTex portal; www.gtexportal.org/home/, accessed on 8 July 2020; https://genenetwork.nl/bloodeqtlbrowser/, accessed on 8 July 2020) [77]. 

## 3. Results

### 3.1. Overall Associations of Selected SNPs with PCa Risk

All SNPs were in HWE in the control group (*p* > 0.001). Logistic regression analysis adjusted for age showed that carriers of the *IGF2BP2*_rs4402960T/T_, *TCF7L2*_rs12255372T/T_, and *TSPAN8|LGR5*_rs7961581C/C_ genotypes had an increased risk of PCa (*p* = 0.037; 0.005 and 0.024), whereas those carrying the *CDKAL1*_rs7754840C_, *FLJ39370*_rs17044137A_*, FTO*_rs9939609A_*, HNF1B*_rs7501939T_*, HNF1B*_rs757210T_, *JAZF1*_rs10486567A_, *KCNQ1*_rs2237897C_, and *KCNQ1*_rs2237892C_ alleles showed a decreased risk of developing the disease (*p* = 0.022, 0.021, 0.046, 0.030, 0.024, 0.011, 0.041, and 0.0002; Table 3). Although none of the reported associations remained statistically significant after a stringent correction for multiple testing (*p* = 0.00048), the QQ plot showed a pronounced and early deviation of identity line, which confirmed that the effect attributed to SNPs in T2D-related loci was more than expected under the null hypothesis and, therefore, might represent true associations (Figure 2).

The identity line represents the null hypothesis (no significant association between T2D-related SNPs and PCa risk). Early deviation of the identity line might represent true associations.

### 3.2. Meta-Analysis

In order to confirm these potentially interesting associations, we conducted a meta-analysis with GWAS data from two large cohorts (UKBiobank and FinnGen) and previously published genetic association studies. After filtering all studies found in the literature according to selected key words, we found that four case-control studies met the eligibility criteria [20,21,22,71]. The meta-analysis of our data with those from all these studies confirmed that carriers of the *FTO*_rs9939609A_, *HNF1B*_rs7501939T_, *HNF1B*_rs757210T_, *HNF1B*_rs4430796G_, and *JAZF1*_rs10486567A_ alleles had a decreased risk of developing PCa (*p* = 3.70 × 10^−5^, 9.39 × 10^−54^, 5.04 × 10^−54^, 1.19 × 10^−71^, and 1.66 × 10^−18^; Table 4). Although the effect of the *FTO* and *HNF1B* loci on PCA risk has been consistently validated in previous studies, this is the first validation study confirming the association of the *JAZF1* variant with the risk of developing the disease. In addition, although the association did not remain significant after correction for multiple testing, the meta-analysis suggested modest associations with the risk of developing the disease for SNPs within the *NOTCH2* and *RBMS1* loci (*p* = 8.49 × 10^−4^ and 0.004; Table 4). These associations are potentially interesting and need to be further investigated.

### 3.3. Functional Characterization of T2D-Related Variants in the HFGP Cohort

In order to test the possible functional relevance of the most interesting SNPs, we analyzed data from the HFGP cohort. The proteomic analysis of the immunological serum proteins showed that carriers of the *HNF1B*_rs7501939T_ allele had decreased circulating levels of ST1A1 protein (*p* = 0.00030; Figure 3). Although this correlation did not survive multiple testing correction, this result supported the implication of the *HNF1B* locus in modulating PCa risk, likely by the regulation of the sulfatation of multiple compounds in the liver. In addition, given the healthy nature of the subjects included in the HFGP cohort, this result is still speculative and needs to be further confirmed in tumor samples of PCa patients. No significant correlation was found between *HFN1B*, *FTO*, *JAZF1*, and *NOTCH2* SNPs and blood-derived cells populations, steroid hormones, or cQTL data, which suggested that these SNPs might affect PCa risk likely through the regulation of mRNA levels of their respective genes.

Next, we assessed functional information from the HaploReg SNP annotation tool, and we also assessed whether any of the potentially interesting markers correlated with mRNA expression levels of their respective genes using data from public eQTL browsers. We found that, according to Haploreg data, the *HNF1B*_rs7501939_, *HNF1B*_rs757210_, and *HNF1B*_rs4430796_ SNPs were modestly associated with mRNA *HNF1B* expression levels in peripheral whole blood (*p* = 9.23 × 10^−4^, 1.00 × 10^−3^, and 2.00 × 10^−3^) [77], and that they mapped among histone marks (H3K4me1, H3K4me3) in several tissues and changed motifs for CEBPB, DMRT5, p300, HES1, and Maf (Appendix A). Similarly, we found that the *NOTCH2*_rs10923931_ and *RBMS1*_rs7593730_ SNPs also correlated with *NOTCH2* and *RBMS1* mRNA expression levels in peripheral whole blood (*p* = 7.3 × 10^−6^ and 3.31 × 10^−7^, respectively) [77]. On the other hand, we found that the *FTO*_rs9939609_ and *JAZF1*_rs10486567_ SNPs mapped among histone marks in multiple tissues and several immune cells, and changed regulatory motifs for multiple regulatory transcription factors (Appendix A). In addition, GTEx portal data suggested that the *NOTCH2*_rs10923931_ SNP is an eQTL in the pancreas and liver.

## 4. Discussion

T2D has been consistently identified as protective factor for PCa development and disease progression [7,8,9]. Several studies have also suggested that both diseases might share a genetic component [13,14,15,16], and some others have attempted to demonstrate the impact of diabetogenic variants on PCa risk, showing controversial results [20,21,22,23]. With this background, we decided to further investigate the association of diabetogenic variants identified through GWAS with the risk to PCa, and attempted to identify the biological mechanisms underlying the most interesting associations through the analysis of functional data from the HFGP cohort and eQTL browsers.

The meta-analysis of the Spanish cohort with those from the UKBiobank, FinnGen, and previously published studies [20,21,22] confirmed that carriers of the *FTO*_rs9939609A_, *HNF1B*_rs7501939T_, *HNF1B*_rs757210T_, *HNF1B*_rs4430796G_, and *JAZF1*_rs10486567A_ alleles showed a decreased risk of developing the disease. Although the association did not reach the stringent significance threshold, we also found that the *NOTCH2*_rs10923931_ and *RBMS1*_rs7593730_ SNPs associated with the risk of developing the disease. The strongest effect on PCa risk was observed for SNPs within the *HNF1B* locus (rs757210, rs7501939, and rs4430796), which showed a similar direction across all study populations. The *HNF1B* (TCF2) gene is located at chromosome 17q12, and it encodes for a transcription factor implicated in the control of regulatory networks related to pancreas and kidney development. It has been reported that the *HNF1B* locus participates not only in the generation of endocrine precursors, but also in the modulation of acinar cell identity and duct morphogenesis. In addition to these functions, it has been consistently reported that the *HNF1B* locus plays a key role in modulating tumorigenesis in solid [79] and hematological cancers [80], and that its methylation or mRNA expression levels can be used for patient stratification [81] and prediction of disease outcome [82]. The association of the *HNF1B*_rs7501939_, *HNF1B*_rs757210_, and *HNF1B*_rs4430796_ polymorphisms with PCa risk was in agreement with results recently reported in GWAS for PCa [14,83,84], whereas large-scale fine mapping studies have even found additional polymorphisms that might contribute to the development of PCa [71]. These findings, together with our functional results reporting that carriers of the *HNF1B*_rs7501939T_ and *HNF1B*_rs757210T_ alleles showed decreased levels of SULT1A1 protein (also known as ST1A1), suggest that the effect of the *HFN1B* locus on PCa risk might be mediated through the regulation of SULT1A1 expression levels. This protein is an enzyme that catalyzes the sulfate conjugation of many hormones, neurotransmitters, drugs, and xenobiotic compounds, among other compounds. It also has been demonstrated that SULT1A1 regulates the metabolic activation of carcinogenic N-hydroxyarylamines, leading to highly reactive intermediates capable of forming DNA adducts, which could result in mutagenesis [85]. In support of the hypothesis of a tumorigenic effect of the SULT1A1, several studies have shown that an increased expression of the SULT1A1 mRNA expression levels contributes to PCa development [86,87]. Although our functional experiments were conducted in a cohort of healthy donors and, therefore, cannot be directly translated to a disease context, our experimental data are in agreement with previous studies, and suggest that the protective effect attributed to the *HNF1B*_rs7501939_ and *HNF1B*_rs757210_ SNPs could be mediated by a reduction in the expression of the SULT1A1 protein. Furthermore, it has been reported that the *HNF1B*_rs7501939_, *HNF1B*_rs757210_, and *HNF1B*_rs4430796_ SNPs are modestly associated with mRNA *HNF1B* expression levels in peripheral blood, which might help to explain how these genetic variants may influence the risk of developing PCa [77]. However, despite these interesting data, we think that the biological link between the *HNF1B* locus and SULT1A1 is still speculative and needs to be further explored and validated, since, if confirmed, it might represent a potentially interesting therapeutic target. An option to confirm this hypothesis would be to measure SULT1A1 levels in tumoral tissues.

Besides these results, this study also confirmed the association of the *FTO* locus with PCa risk. The *FTO* gene is located on chromosome 16q12.2, and it has been implicated in determining not only obesity, but also other symptoms of the metabolic syndrome. In addition, it has been reported that the *FTO* gene acts as a tumor suppressor gene by regulating the proliferation, migration, and invasion of PCa cells, and the *FTO* expression level had a relevance with the development of PCa and the prognosis of PCa patients [88]. Although the association of the *FTO*_rs9939609_ polymorphism with PCa risk was weak in all previous studies, and might depend on different confounding factors, the meta-analysis performed in this study confirmed a strong and consistent association of this intronic variant with a decreased risk of PCa. A recent study also demonstrated that the association of the *FTO*_rs9939609_ SNP with a decreased risk of PCa was found in non-European populations, and that the presence of this genetic marker tended to be associated with disease severity in patients that were overweighted [89]. These findings, together with the lack of significant results in our functional studies, suggest that the role of the *FTO* locus in determining PCa might be mediated by complex obesogenic and/or diabetogenic mechanisms. In support of this hypothesis, we found that the association of the *FTO*_rs9939609_ SNP with a decreased risk of PCa showed a similar direction to the one observed in the GWAS for T2D.

Similarly, we also found that the presence of the *JAZF1*_rs10486567_ SNP was inversely associated with the risk of developing PCa. These results were again in concordance with previous GWAS that have consistently reported that *JAZF1* is a susceptibility locus for PCa [84,90,91,92,93]. The JAZF1 gene is located at 7p15, and it encodes for a zinc finger protein that is overexpressed in the human prostate tissue where it induces cell proliferation, migration, and invasion, and tumor development [94]. Even though there is not much evidence about the functional role of the *JAZF1*_rs10486567_ SNP in PCa, it has been demonstrated that the deletion of the JAZF1 locus is associated with reduced levels of IGF-1 and insulin resistance in mice [95], which suggests that the presence of functional polymorphisms within this locus might act to promote PCa development through diabetogenic mechanisms.

Finally, although the association was not statistically significant after correction for multiple testing, it seems to be reasonable to suggest that genetic variants within the *NOTCH2* and *RBMS1* genes could weakly influence the risk of developing PCa. In this regard, it has been reported that genes of the *NOTCH* family play a relevant role in multiple cancers, including PCa, and that their deregulation may be a key event in tumor onset and disease progression [96,97,98,99,100,101,102]. The human *NOTCH2* locus is located in the chromosomal region 1p11.2, and it plays an important role in modulating prostate development and homeostasis, and its deregulation induces proliferation and expansion of both basal and luminal cells in the prostate [97]. In addition, it has been reported that NOTCH activity promotes prostate cancer cell migration [97], invasion [96,97], aggressiveness [98], and metastasis [99], and that its silencing induces apoptosis and increases the chemosensitivity of PCa cells [100]. However, a tumor suppressive role of the NOTCH pathway has also been suggested in the literature [103,104], which points towards the need of additional studies to elucidate the role of these genes in the etiopathogenesis of PCa. On the other hand, the meta-analysis of all study cohorts suggested that carriers of the *RBMS1*_rs7593730T_ allele had an increased risk of developing PCa. The *RBMS1* gene is located at chromosome 2q24.2, and it encodes for a protein that binds single stranded DNA/RNA and plays an important role in DNA replication, cell cycle progression, gene transcription, and apoptosis. A recent study demonstrated that the *RBMS1* locus acts by regulating the expression of the miR-106b [105], which has been found overexpressed in hepatocellular carcinoma [106], cervical cancer [107], renal carcinoma [108], and gastric cancer [109]. At the same time, Dankert and collaborators have also demostrated that miR-106b can regulate endogenous *RBMS1* expression in PCa cell lines and, thereby, act as a tumor suppressor gene with inhibitory effects on colony formation and cell growth [105]. Despite the lack of evidence linking *RBMS1* SNPs and risk of PCa, it seems to be plausible to suggest that the presence of genetic variants in the *RBMS1* locus might control miR-106b levels and, therefore, favors tumorigenesis. In support of this hypothesis, haploreg data showed that the *RBMS1*_rs7593730_ SNP is associated with different mRNA *RBMS1* expression levels in several tissues and cells [77], and that it maps among histone marks in multiple tissues and several immune cells, and changed regulatory motifs for multiple regulatory transcription factors. Nonetheless, although interesting, the effect of *NOTCH2* and *RBMS1* SNPs on PCa risk must be considered as preliminary and, therefore, needs to be further confirmed in independent cohorts.

This study has both strengths and weaknesses. The major strength of this study is the large number of genetic markers analyzed that allowed us to perform a well-powered meta-analysis of our data with those from previous studies. The meta-analysis of all study cohorts allowed us to not only confirm previous associations between T2D-related polymorphisms and PCa risk, but also to identify potentially interesting genetic markers for the disease. Although the discovery population was relatively small and the influence of diabetogenic variants on the risk of the disease was expected to be modest, our study was sufficiently powered to detect such small effects. Based on the genotype frequencies observed in our study cohort, we had 80% of power (log-additive model) to detect an odds ratio of 1.59 at alpha = 0.00048 (multiple testing threshold) for a polymorphism with a minor allele frequency of 0.25. Likewise, we comprehensively analyzed the impact of T2D-related SNPs in modulating immune responses, blood cell counts, steroid hormones, and serum and plasma metabolites in a relatively large cohort of healthy subjects. However, we also have important limitations. One of them was the fact that functional characterization of the most interesting SNPs was conducted in a healthy control cohort rather than in PCa patients. In addition, we did not have access to medication history, T2D status, and BMI for a substantial number of PCa cases included in the meta-analyses, which did not allow us to adjust our analyses for these confounding variables and, consequently, to rule out the possibility that some of the reported associations could arise as a result of a different distributions of diabetic and/or obese subjects between PCa cases and controls, or because of the effect of diabetes medication rather than the diabetes condition itself. Nonetheless, previous studies have reported that the effect of T2D-related variants on the risk of PCa was independent of T2D status and BMI.

## 5. Conclusions

In conclusion, our study indicates that T2D-related variants within the *HNF1B*, *FTO*, and *JAZF1* genes influence the risk of PCa likely through the modulation of diabetogenic pathways, and suggests, for the first time, an association of SNPs within the *NOTCH2* and *RBMS1* loci that need to be validated in independent cohorts. This study also suggests that the effect of the *HFN1B* SNPs on PCa risk might be mediated by not only the ST1A1 protein, but also *HFN1B* mRNA expression levels, whereas the effect of the *FTO*, *JAZF1*, *NOTCH2*, and *RBMS1* SNPs on PCa risk seem to be involved in the regulation of mRNA expression levels of their respective genes.

## Figures and Tables

**Figure 1 cancers-14-02376-f001:**
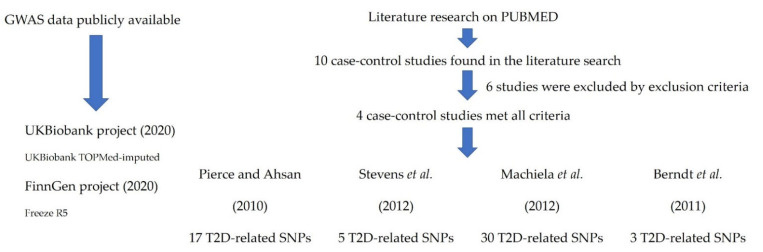
Flow diagram of the study [20,21,22,71].

**Figure 2 cancers-14-02376-f002:**
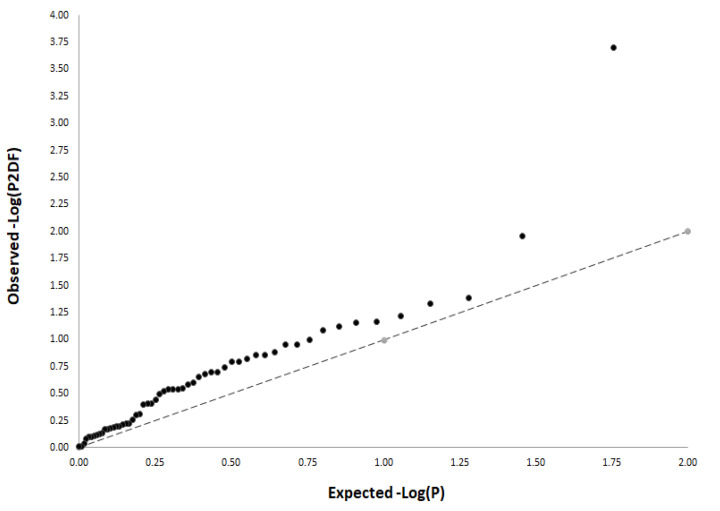
QQ plot showing early deviation of the identity line.

**Figure 3 cancers-14-02376-f003:**
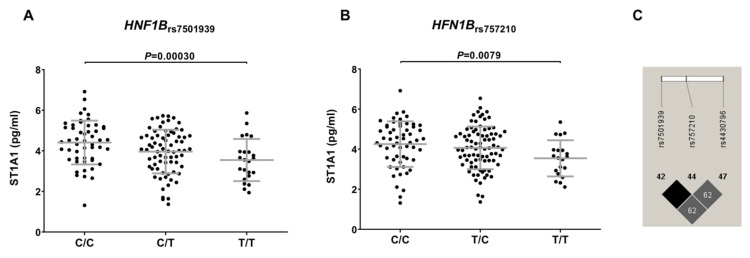
Correlation of the *HFN1B*_rs7501939_ (**A**) and *HFN1B*_rs757210_ (**B**) polymorphisms with reduced levels of ST1A1 protein and linkage disequilibrium values among the *HFN1B* SNPs included in the study (**C**). C/C, cytosine/cytosine; C/T, cytosine/thymine; T/T, thymine/thymine; T/C, thymine/cytosine.

**Table 1 cancers-14-02376-t001:** Demographic and clinical characteristics of PCa patients.

Demographic Characteristics		Study Population (*n* = 990)
Age (years)		62.35 ± 11.51
**Clinical assessment**		
PSA	PSA (4–10)	137 (46.13)
	PSA (10–20)	68 (22.90)
	PSA (>20)	92 (30.97)
Gleason	Gleason (≤7)	220 (73.58)
	Gleason (8–10)	79 (26.42)
TNM Staging system	T1–T2	209 (76.28)
	T3–T4	65 (23.72)
Risk	High	63 (26.58)
	Intermediate	79 (33.33)
	Low	95 (40.09)

**Table 2 cancers-14-02376-t002:** Selected type 2 diabetes-related SNPs.

Gene Name	dbSNP rs#	Nucleotide Substitution	GWAS-Identified Risk Allele for T2D	Location/Aa Substitution	References
ADAM30	rs2641348 ɱ	T/C	C	L359P	[25,26]
ADAMTS9	rs4607103	T/C	C	Near gene	[26,27,28]
ADCY5	rs11708067	T/C	T	Intronic	[29,30]
ADRA2A	rs10885122	G/T	G	Near ADRA2A	[29]
ARAPI, CENTD2	rs1552224	C/A	A	Near gene	[31,32]
CDC123	rs12779790	A/G	G	Near gene	[26,27,28]
CDKAL1	rs7754840	C/G	C	Intronic	[33,34,35]
CDKN2A-2B	rs10811661	T/C	T	Near gene	[26,27,28,33,35,36,37]
COL5A1	rs4240702	C/T	n/s	Intronic	[38]
CRY2	rs11605924	A/C	A	Intronic	[29]
DCD	rs1153188	A/T	A	Near gene	[26]
EXT2	rs1113132	C/G	C	Intronic	[34,39]
FADS1	rs174550	C/T	T	Intronic	[29]
FAM148B	rs11071657	A/G	A	Near gene	[29,40]
FLJ39370	rs17044137	A/T	A	Near gene	[33]
FTO	rs9939609	A/C	A	Intronic	[27,41,42]
G6PC2	rs560887	G/A	G	Intronic	[29,38,43,44,45]
GCK	rs1799884	G/A	A	Near gene	[29,38,43,44,45]
GCKR	rs1260326	A/G	A	Leu446Pro	[46,47,48,49]
HHEX	rs1111875	G/A	C	Near gene	[27,33,34,35,39,41,42]
HMGA2	rs1531343	C/G	C	Near gene	[31,32]
HNF1A, TCF1	rs7957197	A/T	T	Intronic	[31,32]
HNF1B, TCF2	rs7501939 ʯ	C/T	T	Intronic	[14,50]
HNF1B, TCF2	rs757210	C/T	T	Intronic	[14,31]
HNF1B, TCF2	rs4430796	G/A	G	Intronic	[14]
IGF1	rs35767	C/T	C	Near gene	[29,51]
IGF2BP2	rs4402960	G/T	T	Intronic	[27,33,34,35,42,52]
IL13	rs20541	C/T	T	R144Q	[33]
IRS1	rs2943641	C/T	C	Near gene	[31,52,53]
JAZF1	rs864745	T/C	T	Intronic	[26,28]
JAZF1	rs10486567	A/G	A	Intronic	[26,28]
KCNJ11	rs5215	T/C	C	V337I	[27,33,35,41,42,54,55]
KCNJ11	rs5219 ʚ	C/T	T	K23E	
KCNQ1	rs2237897 ʠ	C/T	C	Intronic	[36,56,57,58]
KCNQ1	rs2074196	G/T	G	Intronic	
KCNQ1	rs2237892	C/T	C	Intronic	
KCNQ1	rs2237895	A/C	C	Intronic	
KCNQ1OT1	rs231362	C/T	G	Intronic	[31,32,57]
LTA	rs1041981	A/C	A	T60N	[59]
MADD	rs7944584	A/T	A	Intronic	[29]
MCR4	rs12970134	A/G	A	Near gene	[40]
MTNR1B	rs1387153	C/T	T	Near gene	[31,38,45]
NOTCH2	rs10923931	G/T	T	Intronic	[26,27]
PKN2	rs6698181	C/T	T	Intergenic	[33]
PPARG	rs1801282	C/G	C	P12A	[26,27,33,35,41,42,54,60]
PRC1	rs8042680	A/C	A	Intronic	[31,32]
PROX1	rs340874	A/G	G	Promoter	[29]
RBMS1	rs7593730	T/C	T	Intronic	[61,62]
SLC2A2	rs11920090	A/T	T	Intronic	[29]
SLC30A8	rs13266634	C/T	C	R325W	[27,28,29,33,34,35,39,41,42,63]
TCF7L2	rs7903146 ʞ	C/T	T	Intronic	[27,29,30,33,34,35,39,41,42,63,64,65]
TCF7L2	rs12255372	G/T	T	Intronic	[66]
THADA	rs7578597	T/C	C	Thr1187Ala	[26,67]
TP53INP1	rs896854	T/C	A	Intronic	[31,47,67,68]
TSPAN8, LGR5	rs7961581	C/T	C	Near gene	[69]
VEGFA	rs9472138	C/T	T	Near gene	[26]
WFS1	rs10010131	A/G	G	Intronic	[50]

n/s, not specified; Aa, amino acid; GWAS, genome-wide association studies. ɱ That SNP rs2641348 is in complete linkage disequilibrium with the rs10923931, r^2^ = 1.00. ʞ That SNP rs7903146 is in strong linkage disequilibrium with the rs12255372, r^2^ = 0.72. ʠ That SNP rs2237897 is in strong linkage disequilibrium with the rs2237892, r^2^ = 0.79. ʚ That SNP rs5219 is in complete linkage disequilibrium with the rs5215, r^2^ = 1.00. ʯ That SNP rs7501939 is in strong linkage disequilibrium with the rs4430796, r^2^ = 0.77.

**Table 3 cancers-14-02376-t003:** Association of T2D-related variants and risk of developing PCa in the discovery population.

Variant_dbSNP	Gene	Nucleotide Substitution	Risk Allele	OR (95% CI) ^†^	p
rs2641348	*ADAM30*	T/C	C	0.93 (0.66–1.29)	0.66
rs4607103	*ADAMTS9*	T/C	C	1.06 (0.83–1.37)	0.63
rs11708067	*ADCY5*	A/G	G	1.08 (0.80–1.48)	0.60
rs10885122	*ADRA2A*	G/T	T	1.12 (0.81–1.55)	0.49
rs1552224	*ARAPI, CENTD2*	C/A	A	1.04 (0.76–1.41)	0.82
rs12779790	*CDC123, CAMK1D*	A/G	G	1.05 (0.80–1.38)	0.73
rs7754840	*CDKAL1*	C/G	C	0.69 (0.51–0.95) ^¥^	0.022
rs10811661	*CDKN2A-2B*	T/C	T	0.84 (0.64–1.10)	0.22
rs4240702	*COL5A1*	C/T	T	0.82 (0.66–1.02)	0.082
rs11605924	*CRY2*	A/C	A	1.03 (0.83–1.28)	0.79
rs1153188	*DCD*	A/T	T	0.93 (0.73–1.18)	0.55
rs1113132	*EXT2*	C/G	C	1.02 (0.80–1.30)	0.13
rs174550	*FADS1*	C/T	C	0.96 (0.76–1.22)	0.75
rs11071657	*FAM148B*	A/G	G	1.16 (0.94–1.44)	0.16
rs17044137	*FLJ39370*	A/T	A	0.68 (0.49–0.94) ^¥^	0.021
rs9939609	*FTO*	A/C	A	0.80 (0.63–0.99)	0.046
rs560887	*G6PC2*	G/A	G	1.15 (0.90–1.46)	0.28
rs1799884	*GCK*	G/A	A	1.07 (0.80–1.44)	0.65
rs1260326	*GCKR*	C/T	T	0.93 (0.73–1.20)	0.60
rs1111875	*HHEX*	C/T	C	0.90 (0.72–1.13)	0.36
rs1531343	*HMGA2*	C/G	C	0.74 (0.53–1.02)	0.068
rs7957197	*HNF1A (TCF1)*	A/T	T	0.82 (0.63–1.07)	0.16
rs7501939	*HNF1B (TCF2)*	C/T	T	0.70 (0.50–0.96) ^¥^	0.030
rs757210	*HNF1B (TCF2)*	C/T	T	0.67 (0.48–0.95) ^¥^	0.024
rs4430796	*HNF1B (TCF2)*	G/A	G	0.73 (0.50–1.06) ^¥^	0.10
rs35767	*IGF1*	C/T	C	0.87 (0.66–1.14)	0.30
rs4402960	*IGF2BP2*	G/T	T	1.66 (1.03–2.68) ^§^	0.037
rs20541	*IL13*	C/T	T	0.82 (0.60–1.11)	0.20
rs2943641	*IRS1*	C/T	C	0.97 (0.77–1.21)	0.80
rs864745	*JAZF1*	T/C	T	1.05 (0.84–1.30)	0.67
rs10486567	*JAZF1*	A/G	A	0.69 (0.52–0.91)	0.011
rs5215	*KCNJ11*	T/C	C	0.87 (0.70–1.08)	0.21
rs5219	*KCNJ11*	C/T	T	0.89 (0.71–1.11)	0.29
rs2237897	*KCNQ1*	C/T	C	0.66 (0.44–0.98)	0.041
rs2074196	*KCNQ1*	G/T	T	0.99 (0.53–1.84)	0.97
rs2237892	*KCNQ1*	C/T	C	0.41 (0.26–0.66)	0.0002
rs2237895	*KCNQ1*	A/C	C	0.92 (0.73–1.17)	0.50
rs231362	*KCNQ1OT1*	C/T	C	0.94 (0.75–1.18)	0.61
rs1041981	*LTA*	A/C	A	0.87 (0.68–1.12)	0.29
rs7944584	*MADD*	A/T	T	1.16 (0.93–1.46)	0.18
rs12970134	*MCR4*	A/G	A	0.85 (0.66–1.11)	0.25
rs1387153	*MTNR1B*	C/T	T	0.81 (0.63–1.04)	0.10
rs10923931	*NOTCH2*	G/T	T	0.92 (0.66–1.28)	0.63
rs6698181	*PKN2*	C/T	T	0.90 (0.72–1.13)	0.39
rs1801282	*PPARG*	C/G	C	0.99 (0.70–1.42)	0.98
rs8042680	*PRC1*	A/C	A	1.10 (0.87–1.37)	0.40
rs340874	*PROX1*	A/G	G	0.89 (0.72–1.10)	0.29
rs7593730	*RBMS1*	C/T	T	0.77 (0.59–1.02)	0.070
rs11920090	*SLC2A2*	A/T	T	0.81 (0.59–1.12)	0.20
rs13266634	*SLC30A8*	C/T	C	0.83 (0.65–1.05)	0.11
rs7903146	*TCF7L2*	C/T	T	1.01 (0.80–1.29)	0.91
rs12255372	*TCF7L2*	G/T	T	1.85 (1.20–2.86) ^§^	0.005
rs7578597	*THADA*	T/C	C	0.93 (0.58–1.49)	0.76
rs896854	*TP53INP1*	G/A	A	0.73 (0.52–1.03) ^¥^	0.070
rs7961581	*TSPAN8, LGR5*	C/T	C	1.72 (1.07–2.76) ^§^	0.024
rs9472138	*VEGFA*	C/T	T	1.04 (0.81–1.32)	0.78
rs10010131	*WFS1*	A/G	G	0.90 (0.72–1.13)	0.39

Abbreviations: OR, odds ratio; CI, confidence interval. Estimates were adjusted for age. *p* < 0.05 in bold. ^†^ Estimates calculated according to a log-additive model of inheritance and adjusted for age. ^¥^ Estimates calculated according to a dominant model of inheritance and adjusted for age. ^§^ Estimates calculated according to a recessive model of inheritance and adjusted for age.

**Table 4 cancers-14-02376-t004:** Meta-analysis of association estimates with previous candidate gene association studies according to a log-additive model of inheritance.

			Study Population (304 PCa Cases and 686 Controls)	UKBiobank (2020) (5993 PCa Cases and 168,999 Controls)	FinnGen (2020) (6311 PCa Cases and 74,685 Controls)	Machiela et al. (2012) (2782 PCa Cases and 4458 Controls)	Pierce and Ahsan (2010) (1230 PCa Cases and 1160 Controls)	Stevens et al. (2010) (2935 PCa Cases and 2932 Controls)	Berndt et al. (2011) (10,272 PCa Cases and 9123 Controls)	Meta-Analysis (29,827 PCa Cases and 262,042 Controls)		
SNP	Gene_SNP	*Risk Allele*	OR (95% CI) ^a^	OR (95% CI) ^a^	OR (95% CI) ^a^	OR (95% CI) ^a^	OR (95% CI) ^a^	OR (95% CI) ^a^	OR (95% CI) ^a^	OR (95% CI) ^a^	*p Value*	*P_Het_*
rs2641348	*ADAM30*	C	0.93 (0.66–1.29)	0.95 (0.89–1.01)	0.96 (0.90–1.02)	-	0.87 (0.71–1.05)	-	-	**0.95 (0.91–0.99)**	**0.020**	0.826
rs4607103	*ADAMTS9*	C	1.07 (0.83–1.37)	1.02 (0.97–1.07)	0.97 (0.93–1.02)	0.99 (0.91–1.08)	0.98 (0.85–1.12) η	-	-	0.99 (0.96–1.02)	0.660	0.641
rs11708067	*ADCY5*	G	1.09 (0.80–1.48)	1.00 (0.96–1.05)	0.99 (0.94–1.05)	**0.91 (0.84–0.99)**	-	-	-	0.98 (0.94–1.02)	0.307	0.217
rs10885122	*ADRA2A*	T	1.12 (0.81–1.54)	0.99 (0.94–1.05)	1.00 (0.94–1.06)	-	-	-	-	1.00 (0.96–1.04)	0.863	0.750
rs1552224	*ARAPI*	A	1.04 (0.76–1.41)	1.00 (0.95–1.06)	1.00 (0.95–1.05)	1.00 (0.91–1.10)	-	-	-	1.00 (0.96–1.03)	0.948	0.951
rs12779790	*CDC123*	G	1.05 (0.80–1.37)	1.04 (0.99–1.09)	0.98 (0.93–1.03)	1.06 (0.97–1.16)	1.03 (0.89–1.19) ξ	-	-	1.02 (0.99–1.05)	0.251	0.441
rs7754840	*CDKAL1*	C	0.80 (0.63–1.02)	1.00 (0.96–1.05)	1.03 (0.98–1.07)	1.04 (0.97–1.13)	1.00 (0.88–1.14) #	-	-	1.01 (0.98–1.05)	0.316	0.283
rs10811661	*CDKN2A-2B*	T	0.84 (0.64–1.10)	1.02 (0.97–1.07)	0.95 (0.90–1.01)	**0.91 (0.83–1.00)**	-	-	-	0.98 (0.94–1.01)	0.168	0.063
rs4240702	*COL5A1*	T	0.83 (0.67–1.03)	1.00 (0.97–1.04)	1.01 (0.97–1.05)	-	-	-	-	1.00 (0.97–1.03)	0.907	0.211
rs11605924	*CRY2*	A	1.03 (0.82–1.28)	1.01 (0.97–1.04)	1.00 (0.95–1.04)	-	-	-	-	1.01 (0.98–1.03)	0.637	0.924
rs1153188	*DCD*	T	0.93 (0.73–1.18)	0.97 (0.93–1.02)	0.98 (0.93–1.03)	-	-	-	-	0.97 (0.94–1.01)	0.122	0.892
rs1113132	*EXT2*	C	0.93 (0.73–1.19)	0.99 (0.95–1.02)	1.01 (0.97–1.06)	-	-	-	-	1.00 (0.97–1.02)	0.890	0.646
rs174550	*FADS1*	C	0.96 (0.76–1.21)	0.98 (0.94–1.02)	0.98 (0.94–1.03)	-	-	-	-	0.98 (0.95–1.01)	0.182	0.985
rs11071657	*FAM148B*	G	1.16 (0.94–1.44)	1.02 (0.98–1.06)	1.01 (0.96–1.05)	-	-	-	-	1.02 (0.99–1.05)	0.227	0.456
rs17044137	*FLJ39370*	A	0.79 (0.61–1.03)	1.02 (0.98–1.07)	**0.94 (0.90–0.99)**	-	-	-	-	0.98 (0.95–1.01)	0.199	0.013
rs9939609	*FTO*	A	**0.80 (0.63–0.99)**	0.96 (0.92–1.00)	0.96 (0.92–1.00)	**0.93 (0.86–1.00)**	**0.87 (0.77–0.98) δ**	0.93 (0.85–1.02) ς	-	**0.95 (0.92–0.97)**	**3.70 × 10^−5^**	0.388
rs560887	*G6PC2*	G	1.15 (0.90–1.46)	1.02 (0.98–1.07)	0.96 (0.92–1.01)	-	-	-	-	1.00 (0.94–1.06)	0.705	0.088
rs1799884	*GCK*	A	1.07 (0.80–1.44)	1.03 (0.98–1.08)	0.99 (0.93–1.06)	1.06 (0.96–1.16) ∂	-	-	-	1.02 (0.99–1.06)	0.220	0.643
rs1260326	*GCKR*	C	1.07 (0.83–1.37)	0.99 (0.96–1.03)	0.98 (0.94–1.02)	0.98 (0.91–1.05) ^∏^	-	-	-	0.99 (0.97–1.00)	0.170	0.889
rs1111875	*HHEX*	C	0.90 (0.72–1.13)	1.01 (0.97–1.05)	1.01 (0.97–1.05)	1.01 (0.94–1.09)	0.98 (0.87–1.10)	-	-	1.01 (0.98–1.03)	0.586	0.876
rs1531343	*HMGA2*	C	0.74 (0.53–1.20)	0.98 (0.92–1.04)	1.01 (0.97–1.05)	0.98 (0.88–1.10)	-	-	-	0.99 (0.94–1.03)	0.534	0.512
rs7957197	*HNF1A*	T	0.82 (0.63–1.07)	1.01 (0.97–1.06)	0.99 (0.94–1.04)	0.96 (0.88–1.05)	-	-	-	0.99 (0.96–1.02)	0.673	0.370
rs7501939	*HNF1B*	T	0.84 (0.67–1.05)	**0.83 (0.80–0.86)**	**0.83 (0.79–0.87)**	-	-	**0.87 (0.80–0.94)**	**0.84 (0.80–0.87)**	**0.84 (0.82–0.86)**	**9.39 × 10^−54^**	0.873
rs757210	*HNF1B*	T	0.84 (0.67–1.04)	**0.84 (0.81–0.88)**	**0.82 (0.79–0.86)**	**0.85 (0.79–0.92)**	-	**0.85 (0.79–0.92)**	**0.84 (0.80–0.88)**	**0.84 (0.82–0.86)**	**5.04 × 10^−54^**	0.902
rs4430796	*HNF1B*	G	0.89 (0.71–1.12)	**0.81 (0.79–0.85)**	**0.82 (0.78–0.85)**	-	**0.87 (0.77–0.97)**	**0.85 (0.79–0.92)**	**0.81 (0.77–0.84)**	**0.82 (0.80–0.84)**	**1.19 × 10^−71^**	0.688
rs35767	*IGF1*	C	0.87 (0.66–1.13)	0.99 (0.94–1.04)	1.01 (0.96–1.06)	-	-	-	-	1.00 (0.96–1.03)	0.901	0.516
rs4402960	*IGF2BP2*	T	1.05 (0.83–1.32)	0.99 (0.95–1.03)	1.00 (0.95–1.04)	1.03 (0.95–1.11)	0.91 (0.81–1.04)	-	-	0.99 (0.97–1.02)	0.733	0.552
rs20541	*IL13*	T	0.82 (0.60–1.11)	0.97 (0.93–1.02)	1.04 (0.99–1.08)	-	-	-	-	1.00 (0.93–1.06)	0.788	0.042
rs2943641	*IRS1*	C	0.97 (0.77–1.21)	1.01 (0.97–1.05)	1.02 (0.98–1.06)	0.95 (0.88–1.02)	-	-	-	1.01 (0.98–1.03)	0.641	0.403
rs864745	*JAZF1*	T	1.05 (0.84–1.30)	1.02 (0.98–1.06)	0.99 (0.95–1.03)	**1.08 (1.01–1.16)**	0.98 (0.87–1.10) ℵ	-	-	1.02 (0.99–1.05)	0.269	0.283
rs10486567	*JAZF1*	A	**0.69 (0.52–0.91)**	**0.87 (0.83–0.91)**	**0.86 (0.82–0.91)**	-	-	**0.86 (0.73–0.94)**	-	**0.86 (0.83–0.89)**	**1.66 × 10^−18^**	0.459
rs5215	*KCNJ11*	C	0.87 (0.70–1.08)	1.02 (0.98–1.06)	0.99 (0.95–1.03)	1.01 (0.94–1.09)	**0.89 (0.78–1.00)**	-	-	0.99 (0.96–1.03)	0.921	0.182
rs5219	*KCNJ11*	T	0.89 (0.71–1.11)	1.02 (0.98–1.06)	0.99 (0.95–1.04)	-	-	-	-	1.00 (0.97–1.04)	0.746	0.349
rs2237897	*KCNQ1*	C	**0.66 (0.44–0.98)**	0.94 (0.86–1.04)	0.98 (0.91–1.06)	-	-	-	-	0.94 (0.86–1.04)	0.136	0.148
rs2074196	*KCNQ1*	T	0.99 (0.53–1.84)	1.03 (0.94–1.14)	0.97 (0.88–1.07)	-	-	-	-	1.00 (0.93–1.07)	0.996	0.693
rs2237892	*KCNQ1*	C	**0.41 (0.26–0.66)**	0.98 (0.91–1.06)	1.02 (0.93–1.12)	**0.85 (0.74–0.98)**	0.88 (0.69–1.12)	-	-	0.89 (0.78–1.02)	0.105	0.001
rs2237895	*KCNQ1*	C	0.92 (0.73–1.16)	0.99 (0.95–1.03)	0.96 (0.92–1.00)	-	-	-	-	0.97 (0.95–1.00)	0.078	0.517
rs231362	*KCNQ1OT1*	C	0.94 (0.75–1.18)	0.99 (0.96–1.03)	1.03 (0.99–1.08)	**0.92 (0.86–0.98)**	-	-	-	0.99 (0.94–1.03)	0.515	0.042
rs1041981	*LTA*	A	0.88 (0.69–1.13)	**0.95 (0.91–0.99)**	0.99 (0.94–1.04)	-	-	-	-	**0.96 (0.93–1.00)**	**0.028**	0.359
rs7944584	*MADD*	T	1.16 (0.93–1.46)	1.03 (0.99–1.07)	1.04 (0.99–1.10)	-	-	-	-	**1.04 (1.00–1.07)**	**0.026**	0.585
rs12970134	*MCR4*	A	0.85 (0.65–1.11)	0.99 (0.95–1.04)	0.99 (0.94–1.04)	-	-	-	-	0.99 (0.95–1.02)	0.466	0.541
rs1387153	*MTNR1B*	T	0.81 (0.63–1.04)	1.02 (0.97–1.06)	0.98 (0.94–1.03)	**1.10 (1.01–1.19) ^ς^**	-	-	-	1.01 (0.96–1.08)	0.517	0.029
rs10923931	*NOTCH2*	T	0.92 (0.66–1.28)	0.95 (0.90–1.01)	0.95 (0.89–1.01)	**0.86 (0.76–0.96)**	0.87 (0.71–1.05) *	-	-	**0.94 (0.90–0.97)**	**8.49 × 10^−4^**	0.552
rs6698181	*PKN2*	T	0.90 (0.72–1.13)	0.99 (0.96–1.03)	1.01 (0.96–1.06)	-	-	-	-	0.99 (0.97–1.02)	0.732	0.551
rs1801282	*PPARG*	C	1.00 (0.70–1.42)	1.01 (0.95–1.06)	1.00 (0.94–1.05)	0.96 (0.87–1.07)	0.88 (0.74–1.04)	-	-	0.99 (0.96–1.03)	0.733	0.596
rs8042680	*PRC1*	A	1.10 (0.87–1.37)	0.99 (0.95–1.03)	**0.96 (0.91–0.99)**	1.04 (0.97–1.12)	-	-	-	0.99 (0.95–1.03)	0.300	0.204
rs340874	*PROX1*	G	0.89 (0.72–1.10)	1.01 (0.97–1.05)	1.00 (0.96–1.05)	1.01 (0.94–1.08)	-	-	-	1.00 (0.98–1.03)	0.758	0.708
rs7593730	*RBMS1*	T	0.77 (0.59–1.02)	**1.07 (1.03–1.12)**	1.03 (0.98–1.09)	-	-	-	-	**1.03 (0.96–1.11)**	**0.004**	0.045
rs11920090	*SLC2A2*	T	0.82 (0.59–1.12)	**0.94 (0.89–0.99)**	0.99 (0.93-1.06)	-	-	-	-	**0.96 (0.92–1.00)**	**0.036**	0.307
rs13266634	*SLC30A8*	C	0.83 (0.65–1.05)	0.99 (0.95–1.03)	1.00 (0.96–1.05)	1.00 (0.93–1.08)	0.97 (0.86–1.11)	-	-	0.99 (0.96–1.02)	0.551	0.659
rs7903146	*TCF7L2*	T	1.01 (0.80–1.29)	1.04 (1.00–1.08)	0.99 (0.94–1.04)	**0.90 (0.83–0.97)**	0.97 (0.85–1.10)	-	-	0.98 (0.93–1.03)	0.872	0.047
rs12255372	*TCF7L2*	T	1.17 (0.94–1.46)	1.02 (0.98–1.06)	0.97 (0.92–1.02)	-	-	-	-	1.00 (0.96–1.06)	0.778	0.123
rs7578597	*THADA*	T	1.08 (0.67–1.72)	1.05 (0.98–1.11)	1.04 (0.95–1.14)	1.03 (0.91–1.16)	1.10 (0.92–1.32) ^Ϯ^	-	-	**1.05 (1.00–1.10)**	**0.044**	0.982
rs896854	*TP53INP1*	T	0.88 (0.71–1.10)	0.99 (0.95–1.02)	0.99 (0.94–1.03)	1.02 (0.95–1.09)	-	-	-	0.99 (0.97–1.02)	0.570	0.615
rs7961581	*TSPAN8*	C	1.19 (0.94–1.51)	0.98 (0.94–1.02)	1.00 (0.95–1.05)	1.05 (0.97–1.13)	1.04 (0.92–1.19) τ	-	-	1.00 (0.97–1.04)	0.924	0.295
rs9472138	*VEGFA*	T	1.04 (0.82–1.33)	1.01 (0.97–1.05)	0.98 (0.94–1.03)	-	-	-	-	1.00 (0.97–1.03)	0.877	0.586
rs10010131	*WFS1*	G	0.90 (0.72–1.13)	0.99 (0.95–1.02)	1.01 (0.97–1.05)	1.00 (0.93–1.07)	-	-	-	1.00 (0.97–1.02)	0.859	0.716

Abbreviations: SNP, single nucleotide polymorphism; OR, odds ratio; CI, confidence interval; C, cytosine; T, thymine; A, adenine; G, guanosine; n/s, not specified. a Estimates calculated according to a log-additive model of inheritance and adjusted for age. Meta-analysis was performed assuming a fixed-effect model. *p* < 0.05 in bold. η Authors report the effect found for the rs4411878 (a SNP in complete linkage disequilibrium with the rs4607103, r^2^ = 0.97). ξ Authors report the effect found for the rs11257655 (a SNP in strong linkage disequilibrium with the rs12779790, r^2^ = 0.82). # Authors report the effect found for the rs7756992 (a SNP in strong linkage disequilibrium with the rs7754840, r^2^ = 0.75). δ Authors report the effect found for the rs8050136 (a SNP in complete linkage disequilibrium with the rs9969309, r^2^ = 0.98). ℵAuthors report the effect found for the rs1635852 (a SNP in complete linkage disequilibrium with the rs864745, r^2^ = 0.98). * Authors report the effect found for the rs2641348 (a SNP in complete linkage disequilibrium with the rs10923931, r^2^ = 1.00). ∂ Authors report the effect found for the rs4607517 (a SNP in complete linkage disequilibrium with the rs1799884, r^2^ = 1.00). ^∏^ Authors report the effect found for the rs780094 (a SNP in complete linkage disequilibrium with the rs1260326, r^2^ = 0.92). ^Ϯ^ Authors report the effect found for the rs13414140 (a SNP in complete linkage disequilibrium with the rs7578597, r^2^ = 1.00). τ Authors report the effect found for the rs1353362 (a SNP in strong linkage disequilibrium with the rs7961581, r^2^ = 0.92). **^ς^** Authors report the effect found for the rs10830963 (a SNP in moderate to high linkage disequilibrium with the rs1387153, r^2^ = 0.67). ς Results from Lewis et al. Plos One 2010; 5: e13485 [78].

## Data Availability

The genotype data used in the present study are available from the corresponding authors upon reasonable request. Functional data used in this project have been meticulously catalogued and archived in the BBMRI-NL data infrastructure (https://hfgp.bbmri.nl/, accessed on 7 July 2020) using the MOLGENIS open-source platform for scientific data [52]. This allows flexible data querying and download, including sufficiently rich metadata and interfaces for machine processing (R statistics, REST API), and using FAIR principles to optimize findability, accessibility, interoperability, and reusability [53,54].

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
