# Peer review of "Type 2 Diabetes-Related Variants Influence the Risk of Developing Prostate Cancer: A Population-Based Case-Control Study and Meta-Analysis"

_cancers, 2022, doi:10.3390/cancers14102376_

Round 1
Reviewer 1 Report
Summary:
Previous work has suggested that Type-2 diabetes as important modulators of prostatic tumorigenesis but the molecular mechanisms are still unclear. The authors of this manuscript investigated the influence of 57 GWAS-identified SNPs variants for Type 2 diabetes in the risk of developing prostate cancer. Initially the meta analysis was performed in 304 caucasian PCa patients and 686 male heathy controls. Several SNPs (CDKAL1rs7754840C, FLJ39370rs17044137A, FTOrs9939609A, HNF1Brs7501939T, 278 HNF1Brs757210T, JAZF1rs10486567A, KCNQ1rs2237897C, and KCNQ1rs2237892C) were found to be potentially associated with the decrease of PCa risk but not statistically significant. To further confirm the potentially interestring associations, the author further examined those 57 Type 2 diabetes associated SNPs in two larger cohorts (UKBiobank and FinnGen) together with 4 published case-control studies. This confirmed the association between FTOrs9939609A, HNF1Brs7501939T, HNF1Brs757210T, HNF1Brs4430796G, and JAZF1rs10486567A and decreased risk of PCa development with statistical significance. The authors attempted to get functional characterization of these SNPs by analyzing the data from HaploReg SNP annotation tool and public eQTL browsers.
The authors observed a potential association between HNF1Brs7501939T and ST1A1 levels in the cirlulating blood that is statistically insignificant. The authors further speculated that ST1A1 might be the downstream mediator of this HNF1B SNP. However, the functional characterization of these SNPs in section 3 are logically unclear and the conclusions are not supported by the results. The link between type-2 diabetes associated SNPsPlease find my detailed comments below.
Major:
- Section 3.3 is misleading in calling it functional characterization. The authors performed SNP annotation with is still meta-analysis of the SNP variants annotations generated from the 1000 Genomes project (heathy samples). As pointed out by the authors themselves in the discussion, those annotations were based on healthy individuals rather than relevant patients. Other than the speculations, the author did not provide experimental evidence.
- The conclusions on HNF1B SNP and ST1A1 axis are purely based on speculations. The observed weak association between this HFN1B variant and circulating ST1A1 does not support a causal relationship. In addition, it is confusing why the authors focusing on the circulating ST1A1 levels instead of ST1A1 levels in tissue where HNF1B is expressed but at reduced level due to the SNP. Experimental evidence is needed to support the claims that NHF1B SNP functions through reduced ST1A1 level.
- The importance of HNF1B expression level referenced in the manuscript are limited to the cancer tissue. However, in the section 3, the author ignores the importance of examining the expression level in the tissue context. This makes the lo
- The overall study is based on statistical inferences. But the authors mostly discussing associations that are statistically insignificant after multiple test correction. Without out experimental validation, the accuracy of such statistically insignificant inference is not clear.
- It is logically unclear why the authors includes many data from blood cells, including eQTL and cytokine production experiment. If there is association between blood cells and risk of PCas, the author should provide necessary reference. Otherwise, the authors should better justify the relevance of examining those expression data in blood cells to investigate its attribution to the risk of PCas.
Minor:
- Line 50-51: “No effect of SNPs within these loci and blood-derived cell populations, host immune responses and steroid hormone levels was found, which suggest a diabetogenic effect of these genes to modulate PCa risk.” Is confusing to understand. Recommend editing the grammar to make it clear
- Line 351, it is better to specify the nature of those histone markers examined (enhancer histone markers)
- Line 372-373, definitions of C/C, C/T, T/T were not given in figure legend
- Inconsistent reference citation format in the discussion section.
Author Response
Reviewer #1: Comments and Suggestions for Authors
Summary:
Previous work has suggested that Type-2 diabetes as important modulators of prostatic tumorigenesis but the molecular mechanisms are still unclear. The authors of this manuscript investigated the influence of 57 GWAS-identified SNPs variants for Type 2 diabetes in the risk of developing prostate cancer. Initially the meta-analysis was performed in 304 Caucasian PCa patients and 686 male heathy controls. Several SNPs (CDKAL1rs7754840C, FLJ39370rs17044137A, FTOrs9939609A, HNF1Brs7501939T, 278 HNF1Brs757210T, JAZF1rs10486567A, KCNQ1rs2237897C, and KCNQ1rs2237892C) were found to be potentially associated with the decrease of PCa risk but not statistically significant. To further confirm the potentially interesting associations, the author further examined these 57 Type-2 diabetes associated SNPs in two larger cohorts (UKBiobank and FinnGen) together with 4 published case-control studies. This confirmed the association between FTOrs9939609A, HNF1Brs7501939T, HNF1Brs757210T, HNF1Brs4430796G, and JAZF1rs10486567A and decreased risk of PCa development with statistical significance. The authors attempted to get functional characterization of these SNPs by analyzing the data from HaploReg SNP annotation tool and public eQTL browsers.
The authors observed a potential association between HNF1Brs7501939T and ST1A1 levels in the circulating blood that is statistically insignificant. The authors further speculated that ST1A1 might be the downstream mediator of this HNF1B SNP. However, the functional characterization of these SNPs in section 3 are logically unclear and the conclusions are not supported by the results. The link between type-2 diabetes associated SNPs.
Please find my detailed comments below.
Major:
Section 3.3 is misleading in calling it functional characterization. The authors performed SNP annotation with is still meta-analysis of the SNP variants annotations generated from the 1000 Genomes project (heathy samples). As pointed out by the authors themselves in the discussion, those annotations were based on healthy individuals rather than relevant patients. Other than the speculations, the author did not provide experimental evidence.
Reply: We apologize for the confusing information provided in the manuscript regarding the functional characterization of the T2D-related variants. As stated in the material and methods section, we have performed real experiments in the context of the Human Functional Genomic project (HFGP) that includes a cohort of 520 healthy subjects. Mechanistically, we evaluated the correlation of the T2D-related SNPs with production of 9 cytokines after in vitro stimulation of whole blood, peripheral mononuclear cells and monocyte-derived macrophages from 172 male healthy subjects with LPS, PHA, Pam3Cys, and Staphylococcus Aureus. In parallel, we also tested the correlation between selected SNPs and circulating concentrations of 103 serum and plasmatic inflammatory proteins, 7 plasma steroid hormones and absolute numbers of 91 blood-derived immune cell populations. All specific protocols, biological materials or chemical reagents are carefully detailed in the material and methods section (lines 210 to 275). However, we have to recognize that the Results section was a bit confusing because instead of starting this section describing our main experimental results in the HFGP cohort, we first reported results from Haploreg and eQTL data browsers. We have now amended this section accordingly (please, check lines 357 to 382).
Nonetheless, we agree with the reviewer that the HFGP cohort was built with healthy controls, which is a limitation of our study. We could not confirm our functional results in experiments performed in patients and, therefore, our results should be considered as preliminary and need to be further replicated. We have added a sentence to the strengths and weaknesses paragraph in the discussion that reads like this: “…..Likewise, we comprehensively analyzed the impact of T2D-related SNPs in modulating immune responses, blood cell counts, steroid hormones and serum and plasma metabolites in a relatively large cohort of healthy subjects. However, we also have important limitations. One of them was the fact that functional characterization of the most interesting SNPs was conducted in a healthy control cohort rather than in PCa patients.” (lines 539 to 551)
The conclusions on HNF1B SNP and ST1A1 axis are purely based on speculations. The observed weak association between this HFN1B variant and circulating ST1A1 does not support a causal relationship. In addition, it is confusing why the authors focusing on the circulating ST1A1 levels instead of ST1A1 levels in tissue where HNF1B is expressed but at reduced level due to the SNP. Experimental evidence is needed to support the claims that NHF1B SNP functions through reduced ST1A1 level.
Reply: Again, we apologize for the lack of clarity in the Material and Methods and Results sections. As reported in the lines 256 to 263, a proteomic analysis was performed in serum and plasma samples from the 500FG cohort. Circulating proteins were measured using the commercially Olink® Inflammation panel (Olink, Sweden) that resulted in the measurement of 103 different biomarkers (listed in the Supplementary Table 2). Proteins levels were expressed on a log2-scale as normalized protein expression values, and normalized using bridging samples to correct for batch variation. Therefore, although the healthy nature of the subjects included in the HFGP cohort, hamper direct translation to the pathogenesis of the PCa, we have found a correlation between the HFN1B variant and circulating levels of the ST1A1 protein.
Although we agree with the reviewer that it would have been interesting to measure this protein in tissue, regretfully, we do not have access to this kind of biological material.
The importance of HNF1B expression level referenced in the manuscript are limited to the cancer tissue. However, in the section 3, the author ignores the importance of examining the expression level in the tissue context.
Reply: As mentioned above, we think that it would have been interesting to measure this protein in tissue, regretfully, we do not have access to this kind of biological material. However, we have added a sentence to the manuscript regarding this issue that reads like this: “However, despite these interesting data, we think that the biological link between the HNF1B locus and SULT1A1 need to be further explored and validated since it might represent a potentially interesting therapeutic target. An option to confirm this hypothesis would be to measure SULT1A1 levels in tumoral tissues”.
The overall study is based on statistical inferences. But the authors mostly discussing associations that are statistically insignificant after multiple test correction. Without out experimental validation, the accuracy of such statistically insignificant inference is not clear.
Reply: We disagree with the reviewer. We have performed real experiments that demonstrated the link between T2D-related SNPs and specific proteins that could be considered as preliminary therapeutic targets. Nonetheless, we agree with the reviewer that our experiments should be considered as preliminary and need to be further validated and tested in PCa patients.
It is logically unclear why the authors include many data from blood cells, including eQTL and cytokine production experiment. If there is association between blood cells and risk of PCas, the author should provide necessary reference. Otherwise, the authors should better justify the relevance of examining those expression data in blood cells to investigate its attribution to the risk of PCas.
Reply: The manuscript includes several references regarding the HFGP cohort (References 72 to 74). We have used these data in more recent studies (please check the following papers from our group: Sainz et al. Cancers 2021; Sánchez-Maldonado et al. Frontiers in Immunology 2021).
Minor:
Line 50-51: “No effect of SNPs within these loci and blood-derived cell populations, host immune responses and steroid hormone levels was found, which suggest a diabetogenic effect of these genes to modulate PCa risk.” Is confusing to understand. Recommend editing the grammar to make it clear
Reply: We agree with the reviewer that this sentence was confusing. We have it removed from the Simple Summary section.
Line 351, it is better to specify the nature of those histone markers examined (enhancer histone markers)
Reply: We have added the nature of histone markers examined.
Line 372-373, definitions of C/C, C/T, T/T were not given in figure legend
Reply: Definitions of C/C, C/T, T/T are now provided in the figure legend.
Inconsistent reference citation format in the discussion section.
Reply: We have amended the citation format in the discussion section.
Reviewer 2 Report
First of all, I’d like to congratulate the authors for their tremendous work in evaluating all the data from their research. The article is well written and in a clear and concise manner
Only some minor suggestions:
- Please insert reference number not just the author and year (for example: figure 1 (page 6 line 195), table 4 (page 11, line 325) page 15 line 412,415) not just the author and year in order for the reader to quickly navigate to citation
- A brief explanation of figure 2 is needed. Although an explanation is present at page 9 line 283, in order to increase the readability of the results, the early deviation of the identity line must be briefly explained (example: early deviation of the identity line might represent true association)
- page 10 line 313: “ 4 case-control studies met the eligibility criteria 313 [20-22]” – only 3 references are inserted here for 4 studies. Please insert the reference of the missing study
Author Response
Reviewer #2: Comments and Suggestions for Authors
Summary:
First of all, I’d like to congratulate the authors for their tremendous work in evaluating all the data from their research. The article is well written and in a clear and concise manner
Reply: We thank the reviewer for this positive comment.
Only some minor suggestions:
Please insert reference number not just the author and year (for example: figure 1 (page 6 line 195), table 4 (page 11, line 325) page 15 line 412,415) not just the author and year in order for the reader to quickly navigate to citation
Reply: We agree with the reviewer’s comment. Given that the number of lines would not be present in the final version of the manuscript and page numbers seem not to be a good parameter to find specific citations, it is hard to add that information to figure 1. However, we have added the number of each specific reference to the figure 1, which would undoubtedly help the reader to directly go to any specific report of interest.
A brief explanation of figure 2 is needed. Although an explanation is present at page 9 line 283, in order to increase the readability of the results, the early deviation of the identity line must be briefly explained (example: early deviation of the identity line might represent true association)
Reply: We have added a that reads like this: “The identity line represents the null hypothesis (no significant association between T2D-related SNPs and PCa risk). Early deviation of the identity line might represent true associations”. (Results section: lines 316 to 322.
page 10 line 313: “ 4 case-control studies met the eligibility criteria 313 [20-22]” – only 3 references are inserted here for 4 studies. Please insert the reference of the missing study
Reply: We have added the missing reference.
Reviewer 3 Report
The manuscript Type 2 diabetes-related variants influence the risk of developing prostate cancer: a population-based case-control study and meta-analysis by Sanchez-Maldonado et al, identify the relationship between know SNPs for type 2 diabetes and risk of prostate cancer. The authors also perform a metanalysis using data from this study and previously published studies. Overall, this is an excellent study.
The findings of the protective effects of HNF1B SNPs seem strongest. The functional effects of the HNF1B SNPs through SULT1A1 is also interesting. The new findings of NOTCH2 and RBMS1 SNPs positively associating with prostate cancer are promising but preliminary.
The authors need to report the medication history of diabetic patients. They also need to address if the protective effect of the diabetes associated SNPs (HNF1B) for prostate cancer could potentially be through diabetes medication than the diabetes condition itself.
Author Response
Reviewer #3: Comments and Suggestions for Authors
Summary:
The manuscript Type 2 diabetes-related variants influence the risk of developing prostate cancer: a population-based case-control study and meta-analysis by Sanchez-Maldonado et al, identify the relationship between know SNPs for type 2 diabetes and risk of prostate cancer. The authors also perform a metanalysis using data from this study and previously published studies. Overall, this is an excellent study.
Reply: We thank the reviewer for this positive comment.
The findings of the protective effects of HNF1B SNPs seem strongest. The functional effects of the HNF1B SNPs through SULT1A1 is also interesting. The new findings of NOTCH2 and RBMS1 SNPs positively associating with prostate cancer are promising but preliminary.
Reply: We agree with the reviewer comments. The effect of NOTCH2 and RBMS1 SNPs on PCa risk should be considered as preliminary and, therefore, need to be further validated in independent cohorts. We have added a sentence to the manuscript that reads like this: “Nonetheless, although interesting, the effect of NOTCH2 and RBMS1 SNPs on PCa risk must be considered as preliminary and, therefore, need to be further confirmed in independent cohorts” (Discussion section: lines 526 to 528).
The authors need to report the medication history of diabetic patients. They also need to address if the protective effect of the diabetes associated SNPs (HNF1B) for prostate cancer could potentially be through diabetes medication than the diabetes condition itself.
Reply: We agree with this comment. Regretfully, history of diabetes medication was available only for a small proportion of the subjects included in the study. This information was neither available in the UK Biobank and FinnGen cohorts, which hampered the possibility to comprehensively test whether the effect of T2D-related SNPs on PCa risk was mediated through diabetes medication. However, given that the association of the HNF1B SNPs with PCa has been previously reported and seems to be consistent, we think that the effect of these SNPs is likely mediated by a biological mechanism linked to the diabetic condition rather than a specific diabetic medication that might differ between patients. Nonetheless, in order to clarify this point, we have amended the strengths and weaknesses paragraph (lines 529 to 558) that now reads like this: “In addition, we could not have access to medication history, T2D status and BMI for a substantial number of PCa cases included in the meta-analyses, which did not allow us to adjust our analyses for these confounding variables and, consequently, to rule out the possibility that some of the reported associations could arise as a result of a different distribution of diabetics and/or obese subjects between PCa cases and controls or because the effect of diabetes medication rather than the diabetes condition itself. Nonetheless, previous studies have reported that the effect of T2D-related variants on the risk of PCa was independent on T2D status and BMI.
Round 2
Reviewer 1 Report
The response to the major comments is not convincing and satisfying.
- the "experimental evidence" provided by the authors does not sufficiently support the mechanistic claims. the authors should down-tune the claims on the molecular mechanisms.
- the authors' response on the HFN1B-ST1A1 relationship is not satisfying. The observed correlation does not necessarily indicate its causal relationship. The author should clearly point this out in the manuscript.
- the authors' response on justifying the relevance of examining the expression data in blood cells/plasma to investigate its attribution to the risk of PCas is not satisfying. The previous usage of the public dataset does not necessarily indicate it was used properly in this study
Overall, the story on HNF1B SNP and ST1A1 axis is mostly speculation and is not supported by convincing experimental evidence.
Author Response
1. The "experimental evidence" provided by the authors does not sufficiently support the mechanistic claims. The authors should down-tune the claims on the molecular mechanisms.
Reply: We agree with the reviewer comment. Although we have already mentioned this important limitation in two different paragraphs of the discussion sections (lines 423-426 and 500-501), we have now included additional sentences through the whole manuscript that attempt to down-tune the claims on the molecular mechanisms.
Simple summary: We have added a sentence at the end of the summary pointing out that functional results are speculative and require validation. The sentence is as follows: “However, given the healthy nature of the subjects included in the cohort used for functional experiments, the link between the HNF1B locus and SULT1A1 should be considered still speculative and, therefore, requires further validation”.
Abstract section: We have now mentioned at the end of the abstract that functional mechanisms must be validated in a tissue context. This sentence reads as follows: “These results confirm that functional TD2-related variants influence the risk of developing PCa, but also highlight the need of additional experiments to validate our functional results in a tumoral tissue context”
Results: Section 3.3: We have changed the heading of this section to clarify that experimental studies were conducted in a healthy donor cohort. We have also included a sentence pointing to the limitation of the healthy nature of the subjects included in the HFGP cohort. The sentence reads like this: “In addition, given the healthy nature of the subjects included in the HFGP cohort, this result is still speculative and need to be further confirmed in tumour samples of PCa patients”
Discussion section: We mentioned that our functional results, though interesting, are still speculative. We have amended the following sentence in the discussion section: “we think that the biological link between the HNF1B locus and SULT1A1 is still speculative and need to be further explored and validated since, if confirmed, it might represent a potentially interesting therapeutic target. An option to confirm this hypothesis would be to measure SULT1A1 levels in tumoral tissues” as originally suggested by the reviewer.
2. The authors' response on the HFN1B-ST1A1 relationship is not satisfying. The observed correlation does not necessarily indicate its causal relationship. The author should clearly point this out in the manuscript.
Reply: As mentioned above, we have added multiple specific sentences through the whole manuscript that down-tone the importance of our functional findings in the HFGP cohort.
3. The authors' response on justifying the relevance of examining the expression data in blood cells/plasma to investigate its attribution to the risk of PCas is not satisfying. The previous usage of the public dataset does not necessarily indicate it was used properly in this study.
Overall, the story on HNF1B SNP and ST1A1 axis is mostly speculation and is not supported by convincing experimental evidence.
Reply: We agree with the reviewer that our results are speculative and, therefore, should be considered as preliminary. We also agree with the reviewer that functional experiments should have been conducted in a cohort of PCa patients. However, given the lack of tumoral tissue samples to perform more appropriate functional experiments, our approach represents a feasible option to find unknown links between T2D-related SNPs and specific biological parameters including blood-derived cell counts, serum immunological proteins, cytokines and steroid hormones. Previous outstanding publications have demonstrated that the HFGP cohort is an excellent population to determine the influence of genomic variation on the variability of immune responses (Ter Horst et al. Cell 2016; Orru, V et al. Cell 2013 and Aguirre-Gamboa, R et al. Cell Rep 2016).
